# The Healing Effect of Human Milk Fat Globule-EGF Factor 8 Protein (MFG-E8) in A Rat Model of Parkinson’s Disease

**DOI:** 10.3390/brainsci8090167

**Published:** 2018-08-31

**Authors:** Yoshiki Nakashima, Chika Miyagi-Shiohira, Hirofumi Noguchi, Takeshi Omasa

**Affiliations:** 1Department of Material and Life Science, Graduate School of Engineering, Osaka University, Osaka 565-0871, Japan; omasa@bio.eng.osaka-u.ac.jp; 2Department of Regenerative Medicine, Graduate School of Medicine, University of the Ryukyus, Okinawa 903-0215, Japan; chika@med.u-ryukyu.ac.jp (C.M.-S.); noguchih@med.u-ryukyu.ac.jp (H.N.)

**Keywords:** Parkinson’s disease, dopaminergic neurons, substantia nigra, lipopolysaccharide (LPS), induced pluripotent stem cell (iPSC), milk fat globule-EGF factor 8 protein (MFG-E8)

## Abstract

We searched for drugs that alleviate the reduction of dopaminergic neurons caused by the administration of lipopolysaccharide (LPS) to the substantia nigra of the rat brain. Human milk fat globule-EGF factor 8 protein (MFG-E8) is similar to MFG-E8-S, a short isoform, of the mouse MFG-E8. However, the function of MFG-E8-S was not clear. Rats with LPS-induced Parkinson’s disease were prepared and the effects of human MFG-E8 were examined. MFG-E8 improved the significant reduction in mesencephalic dopamine neurons induced by the administration of LPS. LPS was administered to human induced pluripotent stem cell (iPSC)-derived dopaminergic neurons to induce inflammation and the effect of MFG-E8 was examined. MFG-E8 showed no toxicity toward neurons. We reanalyzed the results using public microarray data. MFG-E8 mRNA was found to be expressed in all parts of the body, particularly by adipose-derived stem cells (ADSCs). Furthermore, we investigated the culture supernatant of ADSCs using the liquid chromatography-tandem mass spectrometry (LC–MS/MS) analysis method and successfully identified the peptide of the MFG-E8 F5/8 type C domain. The results suggested that MFG-E8-S may have a preventive effect against Parkinson’s disease.

## 1. Introduction

Parkinson’s disease (PD) is a progressive neurodegenerative disease; in many cases, the onset occurs in middle age. The incidence is reported to be 100–300 per 100,000 population [1]. The condition is characterized by the reduction of dopamine neurons in the substantia nigra of the midbrain and resultant dopamine deficiency. In recent years, neuroinflammation involving glial cells has been found to be a cause of deterioration in neurodegenerative diseases such as PD. The activation of nerve inflammatory cytokines (e.g., IL-1β, TNF-α, and IL-6) and the microglia by extracellular α-synuclein have been reported to promote the progression of PD [2,3].

It was previously thought that apoptosis due to cytopathogenesis of the neurons plays an important role in the disappearance of dopaminergic neurons, a characteristic pathological condition of PD. However, in the clinical setting, an increase in activated microglia in PD lesions can be visualized by positron emission tomography (PET) using a tracer binding to benzodiazepine receptors and PET [4,5]. Microglia are immune cells that are present in the brain and which carry out phagocytosis. Pathological examinations have revealed the infiltration of microglia and astrocytes into PD lesions, and the levels of inflammatory cytokines (IL-1β, TNF-α, IL-6) are elevated in the affected areas of the brain in PD patients. The microglia and astrocytes are strongly suggested to play a role in the onset of PD [6].

In animal experiments, when inflammatory stimulation is induced by administering lipopolysaccharide (LPS), an outer membrane constituent of gram-negative bacteria, to the rat brain, the levels of inflammatory markers that reflect microglial activity increase before dopamine neuron death [3]. In mice, Toll-like receptor 4 (TLR4), the major receptor of LPS, is more highly expressed in the microglia than in the neurons and astrocytes [7]. Furthermore, the culture supernatant of microglia stimulated with LPS was reported to exert cytotoxic effects when added to the culture solution of nerve cell astrocytes [8]. This result shows that activated microglia play an important role in the molecular pathology of the disappearance of dopamine neurons, one of the symptoms of PD [6].

Multipotent stem cells, such as induced pluripotent stem cells (iPSCs) [9,10], which were discovered by Professor Yamanaka of Kyoto University can induce differentiation into dopamine neurons. Since it was reported last year that the transplantation of iPSC-derived dopamine neurons in a monkey model of PD was successful [11], its application in the treatment of human patients is expected. However, if the condition of PD is evaluated over a long period of time and hyperactive microglia are the cause of PD, the cell death of dopaminergic neurons derived from iPSCs may be induced by activated microglia after transplantation. Thus, it is necessary to understand the mechanism through which microglial activation is controlled in PD patients, and a method for suppressing the microglia of the hyperactivated substantia nigra in PD patients is necessary.

We produced a rat model of PD by administering LPS to the rat substantia nigra. The human MFG-E8 recombinant protein had the effect of alleviating the decrease in dopamine neurons that was induced by the administration of LPS. Thus, in the present study, we aim to investigate the effects of proteins involved in the reduction of dopaminergic neurons in PD and to identify potential candidates for drug discovery.

## 2. Materials and Methods

### 2.1. Reagents

The MSCGM-CD™ Mesencymal Stem Cell Growth Medium BulletKit™ was obtained from Lonza (Basel, Switzerland). hADSCs (46-year-old/female/Caucasian) were obtained from PromoCell (Heidelberg, Germany). Fetal bovine serum (FBS) was obtained from BioWest (Nuaille, France). d-MEM (high glucose) with l-Glutamine, Phenol Red and Sodium Pyruvate (DMEM) was obtained from FUJIFILM Wako Pure Chemical Corporation (Osaka, Japan). Recombinant Human MFG-E8 Protein was obtained from R&D Systems, Inc. (Minneapolis, MN, USA). Lipopolysaccharide (LPS) from *Escherichia coli* O26:B6 was obtained from MilliporeSigma (St. Louis, MO, USA). Plastic dishes were obtained from TPP (Trasadingen, Switzerland). All other materials were of the highest commercial grade.

### 2.2. Animal Care

All experimental protocols were in accordance with the guidelines for the Regulations on Animal Experimentation at Osaka University (Osaka, Japan) and Nihon Bioresearch Inc. (Gifu, Japan). The experimental protocol was approved by the Committee on Animal Experiments of Graduate School of Engineering Osaka University (permit number: 27-5-0); Research Laboratory Center, Faculty of Medicine, and the Institute for Animal Experiments, Faculty of Medicine, University of the Ryukyus (permit number: A2017082).

### 2.3. Preparation of hADSCs

hADSCs (46-year-old Caucasian female) (PromoCell, Heidelberg, Germany) were cultured (37 °C, 5% CO_2_) on a coated 100 mm culture plate (TPP 93100). Then, after reaching 80% confluence passage of cells was performed every 3–4 days. The cells were washed with Phosphate buffered saline (PBS) (calcium, magnesium-free), and hADSCs were dissociated using a dissociation solution. Subculturing was carried out by plating on uncoated 100 mm culture plate. An MSCGM-CD mesenchymal stem cell BulletKit (Lonza 00190632) or DMEM-10% FBS was used for the culture medium. Trypsin/ethylenediaminetetraacetic acid (EDTA) (Lonza CC-3232) was used for the dissociation solution.

### 2.4. Cell Differentiation

Neural stem cell differentiation was performed using PSC Neural Induction Medium (A1647801; Thermo Fisher Scientific, Tokyo, Japan) and a Human Neural Stem Cell Immunocytochemistry Kit (A24354; Thermo Fisher Scientific, Tokyo, Japan) according to the manufacturer’s instructions. Dopaminergic neuron differentiation was performed using a PSC Dopaminergic Neuron Differentiation Kit (A3147701; Thermo Fisher Scientific, Tokyo, Japan) and a Human Dopaminergic Neuron Immunocytochemistry Kit (A29515; Thermo Fisher Scientific, Tokyo, Japan) according to the manufacturer’s instructions.

### 2.5. Preparation of hADSC-CMs for the Analysis of the Protein Expression by LC–MS/MS

The hMSCs used in this study were limited to 3–5 passages in order to match the conditions in which hMSCs are used in the clinical setting. hADSCs were cultured on a 100 mm culture plate using a chemically defined medium (CDM) (MSCGM-CD mesenchymal stem cell BulletKit (Lonza 00190632)) (the number of cells was 3 × 10^6^/plate) until reaching 80% confluence. After 48 h, the culture supernatant was aspirated with a pipette and centrifuged (1500× *g*, 30 min, 4 °C) to remove the cells. After centrifugation of the medium, the supernatant was concentrated 20 times using Amicon Ultra-15, PLGC Ultracell-PL membrane, 10 kDa (MERCK UFC 901008)(Merck Millipore, Darmstadt, Germany), after which the component proteins were analyzed by LC–MS/MS.

### 2.6. Preparation of the Mouse Model of PD

We prepared the mouse model of PD by administering LPS to the rat substantia nigra [12,13,14,15,16,17]. Crl:CD(SD) rats are used. Animals are anesthetized with the intraperitoneal administration (dosing volume: 0.8 mL/kg) of sodium pentobarbital sodium (Tokyo Chemical Industry Co., Ltd., Tokyo, Japan) at 40 mg/kg. After anesthesia, 0.1 mg of levobupivacaine hydrochloride (0.25% poppuscain, manufactured by Maruishi Pharmaceutical Co., Ltd., Osaka, Japan) was administered subcutaneously into the scalp. Next, the hair on top of the animal was trimmed, and the head was fixed to the brain stereotaxic device. After disinfecting the scalp with iodine tincture, an incision is made to expose the skull, the connective tissue on the skull is removed with a cotton swab and dried with a blower to make the position of bregma easier to see. Using a dental drill, a hole is drilled for the insertion of a stainless steel pipe into the skull, at 2.0 mm to the right and 5.3 mm behind the bregma. A silicon tube with an outer diameter of 0.5 mm and a stainless steel pipe connected to a micro syringe are vertically inserted from the bone surface to a depth of 7.8 mm. One microliter of the administered specimen is injected into the substantia nigra over a 5 min period using a micro syringe pump. After injection, the stainless steel pipe is left in place for 5 min, then it is slowly raised and removed. Thereafter, the cranial hole is closed with a non-absorbable myelostatic agent (Nestop, Alfresa Pharma Co., Ltd., Minamigoza, Japan), and the scalp is sutured. The animal is removed from the brain stereotaxic device and returned to the breeding cage. Stainless steel pipes and silicone tubes were sterilized. The doses of the liquid components are as follows: LPS (1 μg/μL) and MFG-E8 (1 μg/μL).

### 2.7. Detection of the TH Expression of Dopaminergic Neurons by Immunohistochemistry

At 14 days after the administration of the drug, the animals were fixed in the dorsal position under anesthesia with sodium pentobarbital (40 mg/kg, administered volume: 1 mL/kg, administered intraperitoneally). The chest was opened to expose the heart and the blood flow of the descending aorta was stopped with forceps. The left ventricle was incised using scissors and the tip of a disposable oral sonde (for mice) connected to a transfusion tube was guided from the left ventricle to the aorta and the oral sonde tip was fixed with a clamp. The atrial appendage was opened, and the animal was euthanized by perfusion with physiological saline and bleeding; 300 mL of saline was then drawn through a transfusion tube. Thereafter, perfusion and fixation was performed with 300 mL of 4% paraformaldehyde (PFA)/PBS solution. After fixation, decapitation was performed, and the skull was removed to remove the whole brain. The brain was placed in a sample bottle containing 4% paraformaldehyde phosphate buffer solution and immersed and fixed under refrigeration. Paraffin embedding was carried out to obtain a block, and tissue slices were prepared according to a conventional method. Rabbit polyclonal Tyrosine Hydroxylase (ab112) was used as the primary antibody. Color development was performed using histofine simple stain rat MAX-PO (MULTI). After nuclear staining a cover slip was applied.

### 2.8. Protein Identification by a Nano LC–MS/MS Analysis

We used an EzRIPA Lysis kit (ATTO Corporation, Tokyo, Japan) for cell lysis according to the manufacturer’s instructions. A protein solution of 2066 μg/mL was obtained from the concentrated solution of culture supernatant. A protein solution of 3541 μg/mL was obtained from the cell lysate of adipose-derived stem cells (ADSC) cultured with CDM. A protein solution of 3800 μg/mL was obtained from the cell lysate of ADSC cultured with DMEM-10% FBS. Finally, 0.4 μg of protein was used for nanoLC–MS/MS. The samples were analyzed with a Nano LC using an UltiMate 3000 RSLC nano system (Thermo Fisher Scientific, Tokyo, Japan) at the Support Center for Advanced Medical Sciences, Institute of Biomedical Sciences, Tokushima University Graduate School by Ikuko Sagawa. The comprehensive expression analysis of proteins using LC–MS/MS was performed according to a previously reported method [18]. In brief, protein-containing solutions were reduced with 10 mM dithiothreitol (DTT)/8 M urea and Tris buffer containing 2 mM EDTA (pH 8.5), alkylated with 25 mM iodoacetamide/8 M Urea and Tris buffer containing 2 mM EDTA (pH 8.5), subsequently diluted with trypsin (pig-derived trypsin) and digested overnight at 37 °C. Peptides were purified and concentrated by solid-phase extraction (SPE) in ZipTip C18 pipette tips (Merck Millipore, Darmstadt, Germany). Nano LC–MS/MS was performed using an UltiMate 3000 RSLC nano system (Thermo Fisher Scientific, Tokyo, Japan). The reconstituted peptides were injected into an Acclaim PepMap C18 trap column (75 μm × 15 cm, 2 μm, C18) (Merck Millipore, Darmstadt, Germany). Solvent A was 0.1% formic acid. Solvent B was 80% acetonitrile/0.08% formic acid. The peptides were eluted in a 229 min gradient of 4% solvent B in solvent A to 90% solvent B in solvent A at 300 nL/min. Orbitrap Elite’s ionization method was set to Nanoflow-LC ESI, positive, and the capillary voltage was set to 1.7 kV. Tandem mass spectrometry was performed using the Proteome Discoverer software program (version 1.4, Thermo Fisher Scientific, Tokyo, Japan). Charge stated deconvolution and deisotoping were not performed.

### 2.9. Data Analyses

#### 2.9.1. Database Searching

All raw data were searched against the SwissProt 2016-07 database using the Mascot 2.5.1 software program (Matrix Science, London, UK) (unknown version, 551,705 entries). The peptide tolerance and MS/MS tolerance were set to 10 ppm and 0.6 Da, respectively. False discovery rates (FDRs) were calculated for each of the samples using the following formula: FDR = (Ndecoy/Nreal + NDecoy) × 100. This indicates the percentage of random or “false” peptide identifications in the raw data. The relative abundance of proteins identified by LC–MS/MS was estimated by determining the protein abundance index (PAI) and the exponentially modified protein abundance index (emPAI). Visualized and validated complex LC–MS/MS proteomics experiments were performed using Scaffold (http://www.proteomesoftware.com/) to compare samples in order to identify biological relevance.

#### 2.9.2. Criteria for Protein Identification

The Scaffold software program (version 4.7.3, Proteome Software Inc., Portland, OR, USA) was used to validate the MS/MS-based peptide and identify proteins. Peptide identifications were accepted if they could be established at >46.0% probability to achieve an FDR of <1.0% using the Scaffold Local FDR algorithm. Protein identifications were accepted if they could be established at >5.0% probability to achieve an FDR of <1.0% and contained at least 2 identified peptides. Protein probabilities were assigned by the Protein Prophet algorithm [19]. Proteins that contained similar peptides and which could not be differentiated based on MS/MS alone were grouped to satisfy the principles of parsimony. Proteins that shared significant peptide evidence were grouped into clusters. A protein GO analysis was performed using the GO analysis function of the Scaffold 4 software program with imported data (goa_uniprot_all.gaf (downloaded 14 October 2016)) from the external GO Annotation Source database.

### 2.10. RefEx Analysis

RefEx (Reference Expression dataset: http://refex.dbcls.jp/) is an information browsing tool visualized by connecting the following four databases. This web-based tool can be used to visually compare and browse large amounts of genomic genetic information, including DNA microarrays [20].

* EST (expressed sequence tag): EST is a short sequence that is used as a “marker” of RNA transcripts. The EST data were obtained from the NCBI UniGene database (https://www.ncbi.nlm.nih.gov/unigene).

* GeneChip: The gene expression database of the Affymetrix™ GeneChip DNA microarray (Thermo Fisher Scientific, Tokyo, Japan) was obtained from the NCBI GEO database (https://www.ncbi.nlm.nih.gov/geo/).

* CAGE: CAGE (Cap Analysis Gene Expression: a technique to capture and sequence the 5 ‘end of a capped mRNA) was obtained from the RIKEN FANTOM 5 project (http://fantom.gsc.riken.jp/5/) mRNA expression database.

* RNA-seq: The RNA mRNA database of RNA sequences measured by the Illumina Genome Analyzer was obtained from NCBI Sequence Read Archive (https://trace.ncbi.nlm.nih.gov/Traces/sra/) or the European Nucleotide Archive (https://www.ebi.ac.uk/ena).

The Human Tyrosine hydroxylase (TH) (Refseq ID: NM_000360) DNA expression levels in 10 major groups of normal tissues (brain, blood, connective tissue, reproductive tissue, muscle, digestive tract, liver, lung, kidney, and urinary tract) were quantitatively analyzed using RefEx (Reference Expression data set; http://refex.dbcls.jp/) based on microarray data sets for humans. Please see (http://refex.dbcls.jp/gene_info.php?lang=ja&db=human&geneID=7054&refseq=NM_000360&unigene=Hs.435609&probe=208291_s_at) for the probes used for TH.

The human MFG-E8 (Refseq ID: NM_001114614) DNA expression levels in 10 major groups of normal tissues (brain, blood, connective tissue, reproductive tissue, muscle, digestive tract, liver, lung, kidney, and urinary tract) were quantitatively analyzed using RefEx (Reference Expression data set; http://refex.dbcls.jp/) based on microarray data sets for humans. Please see (http://refex.dbcls.jp/gene_info.php?lang=ja&db=human&geneID=4240&refseq=NM_001114614&unigene=Hs.374503&probe=210605_s_at) for the probes for MFG-E8.

### 2.11. The BioGSP Analysis

BioGSP (http://biogps.org/#goto=welcome) [21] is a database of gene expression profiles in various tissues and cell lines of humans, mice, and rats using the GeneChip dataset, which is a microarray manufactured by Affymetrix™ (Thermo Fisher Scientific, Tokyo, Japan) Corporation.

Materials and Methods should be described with sufficient details to allow others to replicate and build on published results. Please note that publication of your manuscript implicates that you must make all materials, data, computer code, and protocols associated with the publication available to readers. Please disclose at the submission stage any restrictions on the availability of materials or information. New methods and protocols should be described in detail while well-established methods can be briefly described and appropriately cited.

Research manuscripts reporting large datasets that are deposited in a publicly available database should specify where the data have been deposited and provide the relevant accession numbers. If the accession numbers have not yet been obtained at the time of submission, please state that they will be provided during review. They must be provided prior to publication.

Interventionary studies involving animals or humans, and other studies require ethical approval must list the authority that provided approval and the corresponding ethical approval code.

## 3. Results

### 3.1. Distribution of Systemic DNA and the mRNA Expression of TH

RefEx (Reference Expression dataset) (http://refex.dbcls.jp/) is a web tool for searching and browsing the gene expression data of normal tissues, primary cultured cells, and cell lines obtained by four different experimental methods (EST, GeneChip, CAGE, RNA-seq). The human TH was recognized in Refseq ID (NM_000360), GeneID (7054), Unigene ID (Hs. 435609), and Probe set ID (208291_s_at) codes. Three-dimensional human body models (whole body, internal organs, brain/coronal and brain/sagittal) were created from GeneChip data (Figure 1A, upper panel). The TH expression was generally low in the whole body image, but was moderately expressed in the adrenal gland and brain (Figure 1B, left panel). Within the brain, TH was moderately expressed in the corpus callosum. In the other regions of the brain, the expression levels of TH were low (Figure 1A, upper panel). CAGE data showing the mRNA expression levels revealed that among the body organs, the TH mRNA expression was relatively high in the blood vessels, corpus callosum, and brain stem (Figure 1B, right panel).

### 3.2. Distribution of Systemic DNA and the mRNA Expression of MFG-E8

The human MFG-E8 was recognized in the codes of Refseq ID (NM_001114614), GeneID (4240), Unigene ID (Hs. 374503), Probe set ID (210605_s_at). Human body 3D models (whole body, internal organs, brain/coronal and brain/sagittal) were created from GeneChip data (Figure 1A, lower panel). MFG-E8 protein has previously been reported to be secreted by macrophages. However, the expression of MFG-E8 was observed in all body organs and the high expression of MEF-E8 was observed in blood vessels, but not the heart. The expression of MFG-E8 was uniform throughout the brain (Figure 1B, left panel). The high expression of MFG-E8 mRNA in the systemic organs indicated by CAGE was consistent with the DNA expression of MFG-E8 in the systemic organs (Figure 1B, right panel). It is considered impossible to logically explain why the high expression of MFG-E8 was detected in all body organs if macrophages are the sole cause. This result suggested that MFG-E8 is expressed by cells other than macrophages in all of the organs of the body.

### 3.3. Comparison of the Human MFG-E8 mRNA Expression Levels in the Organs, Brain, and Different Cell Types

The tissue-specific mRNA expression patterns can indicate important clues about genetic functions. The pattern of gene expression can be examined on the genome scale using a high-density oligonucleotide array. The mRNA expression pattern of human MFG-E8 was shown from BIOGPS (biogps.org). Dataset; GeneAtlas U 133 A (http://biogps.org/#goto=genereport&id=4240) (gcrma, species; human; sample No; *n* = 176) [22]. The MFG-E8 mRNA expression ratio in various organs was shown. MFG-E8 was highly expressed in the retina and uterus. Surprisingly, an extremely high expression level was observed in adipocytes (Figure 2A). The expression ratio of MFG-E8 mRNA was mainly shown in hematologic cancer cells. We cannot generalize the findings to all cancer cells; however, MFG-E8 did not tend to be specifically expressed in cancer cells (Figure 2B). The MFG-E8 mRNA expression levels in various parts of the brain are shown. MFG-E8 was expressed throughout the brain. The expression was particularly high in pineal day and pineal night (Figure 2C). However, in the data on blood cells, no types of hematopoietic cells showed high MFG-E8 mRNA expression levels (Figure 2D). Thus, in the pineal day and pineal night, it was suggested that cells that are specifically present in the brain express MFG-E8 mRNA. MFG-E8 was previously reported to be expressed in mouse embryonic fibroblasts (MEF) [23]. On the other hand, the MFG-E8 mRNA expression was low in the human fetal brain, fetal thyroid, fetal lung, fetal liver (Figure 2E). This result indicates that MFG-E8 is not generally highly expressed in each part of the fetus body. Whether or not MFG-E8 is highly expressed in human embryos is unknown; however, the weak expression of MFG-E8 was confirmed. This result shows that MFG-E8 mRNA is generally expressed at the developmental stage.

### 3.4. Human MFG-E8 Prevents the LPS-Induced Reduction of Dopamine Neurons in the Rat Nigra

The effects of LPS and MFG-E8 on nigral dopamine neurons. Each drug was administered into the substantia nigra of rats, and the effects were evaluated by tissue immunostaining method using TH antibodies. TH-positive dopamine neurons were observed in the substantia nigra following the administration of MQ after manipulation treatment (Figure 3A, left panels). The stained area (Figure 3B) and the number of TH-positive dopamine producing cells in the nigra of rats treated with LPS (1 μg/1 μg) (Figure 3C) were significantly lower in comparison to the control group that was treated with MQ. A micrograph of each tissue section of 3 rats is shown (Figure 3A, middle panels). Recombinant human MFG-E8 protein was admixed with LPS and administered to the rat substantia nigra. Although the same amount of LPS was administered, the administration of recombinant human MFG-E8 had a preventive effect against the significant decreases in the area of TH-positive dopamine neurons (Figure 3B) and in the number of TH-positive cells (Figure 3C). A micrograph of each tissue section of 3 rats is shown (Figure 3A, right panel).

### 3.5. Preparations of hADSC-CMs for the Analysis of the Protein Expression by LC–MS/MS

A 2 mL quantity of CDM was added to hADSC (1 × 10^6^ cells) and cultured for 48 h to prepare hADSC-CMs, and was then concentrated to 1/20 of the original volume using a 10 k filter, the component proteins were analyzed by LC–MS/MS. Twentyfold concentrated hADSC-CM was subjected to LC–MS/MS (Figure 4A).

### 3.6. The Peptide Sequence of MFG-E8 Secreted or Expressed by hADSCs

Peptide fragments of MFG-E8 contained in the hADSC-CM sample (Figure 4B), the cell lysate sample of hADSC (CDM) (Figure 4C) and the cell lysate sample of hADSC (DMEM-10% FBS) (Figure 4D) were detected by LC–MS/MS. The site of the peptide sequence, which was identified using the Scaffold 4 software program, was visualized. At 13% coverage of hADSC-CM samples, peptide chains of positions 70–107, 207–219 were detected. At 10% coverage of cell lysate samples of hADSC (CDM), peptide chains of positions 93–107, 111–134 were detected. At 28% coverage of cell lysate samples of hADSC (DMEM-10% FBS), peptide chains of positions 93–107, 111–135, 220–237, 254–268, 272–308 were detected. The peptide chains detected in the hADSC-CM sample were the F5/8 type C1 domain and F5/8 type C2 domain. The peptide chain detected in the hADSC (CDM) cell lysate sample was the F5/8 type C1 domain. The peptide chains detected in the hADSC cell lysate sample (DMEM-10% FBS) were the F5/8 type C1 domain and the F5/8 type C2 domain. The EGF-like domain (a cell attachment site) has a cell adhesive active sequence (RGD motif) at positions 46–48 of the peptide sequence. No peptide sequence of the EGF-like domain was detected in any of the hADSC samples. On the other hand, the F5/8 type C domain binding to phosphatidylserine (PS), which was expressed on the surface of dead cells was detected in all hADSC samples (Figure 4E).

### 3.7. The Biological Processes, Cellular Components, and Molecular Function of Proteins Identified from hADSC-CM, Cell Lysate of hADSC Cultured in CDM or DMEM-10% FBS

The biological processes of proteins were analyzed using the Mascot software program with the SwissProt 2016 database. The human MFG-E8 was listed by GO analysis for protein functions classified into the following categories: “Biological Processes”, “Cellular Components”, and “Molecular Function”.

#### 3.7.1. Biological Processes

The GO analysis revealed that MFG-E8 was associated with the following biological processes: for Biological adhesion, cell adhesion; for Biological regulation, positive regulation of apoptotic cell clearance; for Cellular processes, cellular protein metabolic process, phagocytosis, engulfment, phagocytosis, recognition, and viral process; for Developmental processes, angiogenesis; for Establishment of localization, phagocytosis, engulfment, phagocytosis, and recognition; for Localization, phagocytosis, engulfment, phagocytosis, and recognition; for Metabolic processes, the cellular protein metabolic process; for Multi-organism processes, single fertilization and viral process; for Multicellular organismal processes, angiogenesis; for Reproductive processes, single fertilization; and for Viral processes, viral process.

#### 3.7.2. Cellular Components

The GO analysis revealed that MFG-E8 was associated with the following cellular components: for the Extracellular region, extracellular exosome, extracellular matrix, extracellular space, and extracellular vesicle; for Membrane, external side of plasma membrane, extrinsic component of plasma membrane, and membrane; and for Plasma membrane, external side of plasma membrane and extrinsic component of plasma membrane.

#### 3.7.3. Molecular Function

The GO analysis revealed that MFG-E8 was associated with the following molecular functions: for Binding, integrin binding, phosphatidylethanolamine binding, and phosphatidylserine binding; and for Molecular function, integrin binding, phosphatidylethanolamine binding, and phosphatidylserine binding.

### 3.8. The Effects of LPS and Human MFG-E8 on iPS Cell-Derived Dopamine Neurons In Vitro

The differentiation of iPSC-derived dopamine neurons was induced using a Dopaminergic Neuron Differentiation Kit (Thermo Fisher Scientific, Tokyo, Japan). The differentiation maturity of dopamine neurons was evaluated by fluorescence immunostaining of TH, FOXA2, OTX2 using Human Dopaminergic Neuron Immunocytochemistry Kit (Thermo Fisher Scientific, Tokyo, Japan) (Figure 5A,B). These observations confirmed that the induction of differentiation of iPSC-derived dopamine neurons had occurred normally. We investigated whether LPS induces cell death in hiPSC-derived dopamine neurons in vitro. In both the MTT assay to measure cell viability (Figure 5C, top panel) and the LDH assay to measure cell death (Figure 5C, lower panel), LPS (1 μg/mL) had no effect on the cell viability or the cell death activity of dopaminergic neurons derived from hiPSCs. Furthermore, the co-administration of recombinant human MFG-E8 protein (1 μg/mL) and LPS did not affect the activity of living dopamine neurons derived from hiPSCs or their cell death. In order to investigate whether LPS has an effect on the viability of living hiPS cell-derived dopamine neurons, the effects of different doses of LPS were investigated. LPS was prepared at doses of 1–100 μg/mL and administered to hiPSC-derived dopamine neurons; however, no dose was found to suppress the cellular activity of dopamine neurons derived from hiPSCs (Figure 5D). These results also indicate that MFG-E8 was not cytotoxic to hiPSC-derived dopamine neurons.

### 3.9. The Effects of LPS and Human MFG-E8 on iPS Cell-Derived Neurons In Vitro

The differentiation of iPSC-derived dopamine neurons was induced using PSC Neural Induction Medium (Thermo Fisher Scientific, Tokyo, Japan). Differentiation maturity of neurons was evaluated by fluorescence immunostaining of SOX1, SOX2, PAX6, and NESTIN using a Human Neural Stem Cell Immunocytochemistry Kit (Thermo Fisher Scientific, Tokyo, Japan) (Appendix A). These observations confirmed that the induction of differentiation of iPSC-derived neurons was performed normally. We investigated whether LPS induces cell death in hiPSC-derived neurons in vitro. The MTT assay (Appendix A, top panel) and LDH assay (Appendix A, bottom panel) revealed that LPS (1 μg/mL) had no effect on cell viability activity or cell death, respectively, of neurons derived from hiPSCs. In addition, the co-administration of recombinant human MFG-E8 protein (1 μg/mL) with LPS had no effect on the cell viability or cell death of neurons derived from hiPSCs. These results also showed that MFG-E8 was not cytotoxic to hiPSC-derived neurons.

## 4. Discussion

In the present study, recombinant human MFG-E8 protein was administered to rats. Unlike rodents, human MFG-E8 has a short isoform structure without EGF-like domain 2 and P/T domain [24,25,26,27,28]. In a study of MFG-E8 deficient mice, the mouse-specific long isoform MFG-E8-L (61–72 kDa) was shown to promote the phagocytosis of apoptotic cells and neurons by macrophages [29,30,31] and microglia [32]. However, the short isoform—which lacks phagocytosis-promoting activity (MFG-E8-S; 48–56 kDa)—is extensively expressed in more organs and its role has not been elucidated [26]. Several reports describe MFG-E8 as a multifunctional molecule involved in the regulation of the cell surface of organs; it has been reported to play an important role in maintaining intestinal epithelial homeostasis and promoting mucosal healing [33]. The role of angiogenesis in the retina has also been reported [34,35]. MFG-E8 is also known to be a potent angiogenesis-promoting molecule that is involved in vascular branching [36,37]. The functions of these MFG-E8 were functionally classified by GO analysis. The results were listed in the MFG-E8 dataset under the categories of Biological Processes, Cellular Components, and Molecular Function. These reports are consistent with the 3D image of the distribution of the MFG-E8 expression that was observed in the GeneChip dataset (Figure 1A, bottom panel). However, CAGE indicates that MFG-E8 mRNA is expressed in all organs (Figure 1B, right panel). Thus, it is presumed that MFG-E8 is expressed in a wide range of vascular networks throughout the body. Indeed, it is reported that human MFG-E8 activates the proliferation of vascular smooth muscle cells through integrin signaling in rats. The vascular smooth muscle cell itself is also reported to express MFG-E8 mRNA [38]. On the other hand, CAGE shows that TH mRNA is localized and expressed in the brain stem (Figure 1A, top panel and Figure 1B, right panel). The expression profiles of MFG-E8 and TH mRNA were not correlated.

The tissue- and cell-specific mRNA expression patterns can indicate important clues about genetic function. High-density oligonucleotide arrays allow for gene expression patterns to be examined on a genome scale (Figure 2A–E). Human MFG-E8 mRNA is strongly expressed in the retina, adipocytes and uterus; the expression in adipocytes is especially high (Figure 2A). Human MFG-E8 mRNA is particularly expressed in pineal day and pineal night in the brain (Figure 2C). The data of the RNA-sequence of the database RefEx (http://refex.dbcls.jp) showed that the expression scores of human MFG-E8 were as follows: lymph node, 2.22; fat, 2.92; and mammary gland, 2.19. Among all of the organs in the body, the expression MFG-E8 mRNA was highest in the fat (http://refex.dbcls.jp/gene_info.php?lang=ja&db=human&geneID=4240&refseq=NM_001114614&unigene=Hs.374503&probe=210605_s_at). This finding was consistent with the data obtained using high-density oligonucleotide arrays. The survey on the sites of MFG-E8 expression using these databases revealed that MFG-E8 was highly expressed in adipocytes. Peptide fragments of MFG-E8 were also detected by an expression analysis of the MFG-E8 protein using our hADSCs. This result indicated that the ADSCs contained in fat express MFG-E8 (Figure 4B–D). In our analysis using LC–MS/MS, all MFG-E8 detected from the sample of hADSCs showed the F5/8 type C domain of MFG-E8 (region binding with PS), and no cell attachment site (RGD) was detected (Figure 4B–D).

In LPS-induced Parkinson’s rat models, the number of dopamine neurons is decreased due to the induction of excessive microglial activity around the substantia nigra. The systemic administration of LPS induces brain inflammation but does not induce dopaminergic neuronal death in the substantia nigra [39]. As a possible explanation, it is reported that inflammatory cytokines such as IL-1β, TNF-α, IL-6 are released as a result of microglia-induced neuronal cell death [2,39,40,41]. We co-administered LPS and human MFG-E8 in an in vitro experiment using hiPSC-derived neurons (Appendix A), and further differentiated mature dopamine neurons (Figure 5C). LPS and human MFG-E8 were found to have no effect on cell viability or cell death of hiPSC-derived neurons and dopamine neurons derived from hiPSCs. The co-administration of human MFG-E8 protein and LPS to the rat substantia nigra revealed that it had a preventive effect against the significant reduction in the number of nigral dopamine neurons that was induced by the administration of LPS (Figure 3A–C). This result suggests that the administration of human MFG-E8 to the substantia nigra might have influenced the excessive microglial activity. In multiple studies with mice, MFG-E8 has been reported as an essential factor for neuronal loss of Alzheimer’s disease as a microglial secreted protein [42,43,44]. However, since the macrophages and microglia in mice secrete MFG-E8-L, the use of human MFG-E8 (MFG-E8-S) in this study might have led to conflicting results. Another study reported that MFG-E8 did not participate in nigral dopamine neuronal death in a mouse model of PD [45]. Thus, further studies on MFG-E8-S and MFG-E8-L are required to understand the relationship between MFG-E8 and PD.

Based on the results of this study, we hypothesized that MFG-E8-S contributes to the prevention and cure of PD in vivo through a different mechanism of action from MFG-E8-L type. As a background, we proposed this hypothesis as one explanation for the extensive overexpression of MFG-E8 mRNA in vivo. We are considering whether ADSCs play a role as a major source of MFG-E8-S in vivo. It is also reported that ADSCs, which are therapeutic cells, pass through the blood brain barrier [46]. In addition, MFG-E8-S protein may flow into the bloodstream and possibly affect the brain. Human MFG-E8 (MFG-E8-S), a protein secreted by ADSC, may support the curative effect of dopamine neuron transplantation using iPSCs and may promote the therapeutic effects in patients with PD. It may also inhibit the microglial activity after the transplantation of dopaminergic neurons derived from hiPSCs.

## 5. Conclusions

We examined hADSCs in detail to scientifically examine their effects in cell therapy. As a result, we found that hADSCs are highly likely to be a major source of MFG-E8-S protein in vivo. In this study, the dopamine neurons of a rat model with Parkinson’s disease induced by LPS were shown to be protected by human MFG-E8. This result may indicate that the secretion of MFG-E8-S by hADSCs has a preventive effect against Parkinson’s disease.

## Figures and Tables

**Figure 1 brainsci-08-00167-f001:**
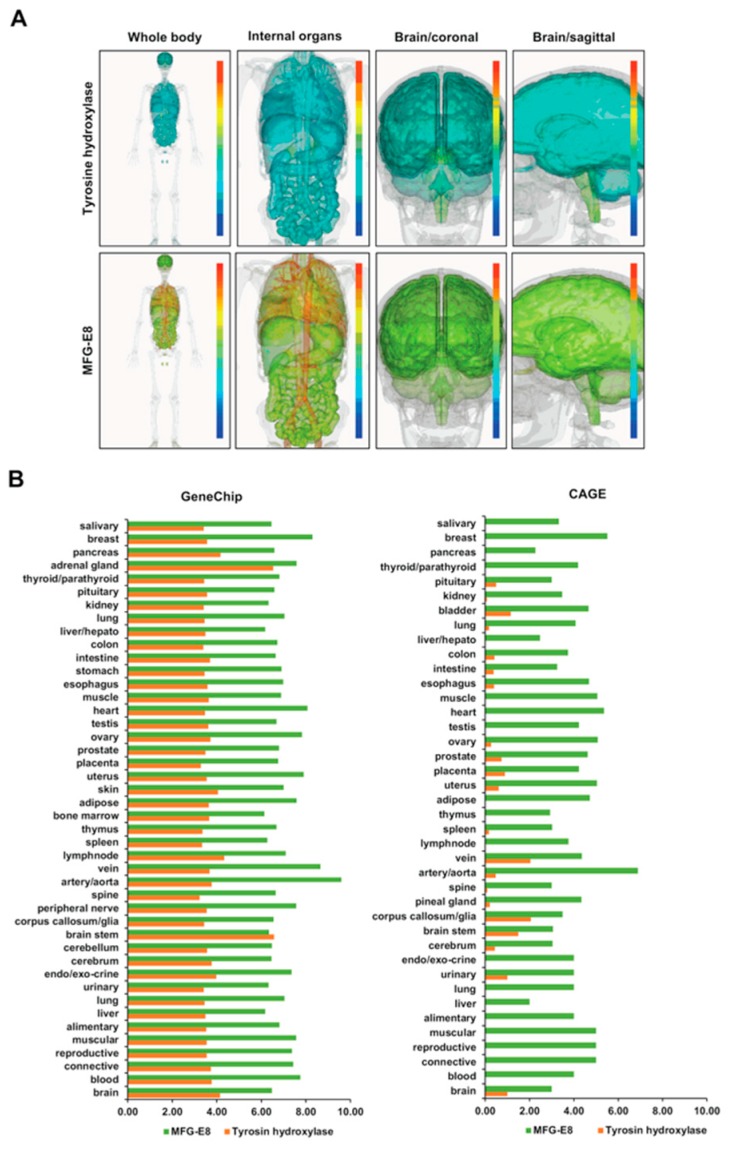
The distribution of the TH and MFG-E8 DNA and mRNA expression by the systemic organs of the whole body. The RefEx-based quantification of the DNA expression of the TH (**A**, upper panel) and MFG-E8 (**A**, lower panel) genes. From the left, models of human transmission in the whole body, internal organs, brain (coronal), brain (sagittal). Red and blue denote high and low expression levels, respectively. The DNA expression of MFG-E8 and TH in 44 major groups of normal tissues (**B**, left panel). The mRNA expression of MFG-E8 and TH was identified in the 44 major groups of normal tissues (**B**, right panel).

**Figure 2 brainsci-08-00167-f002:**
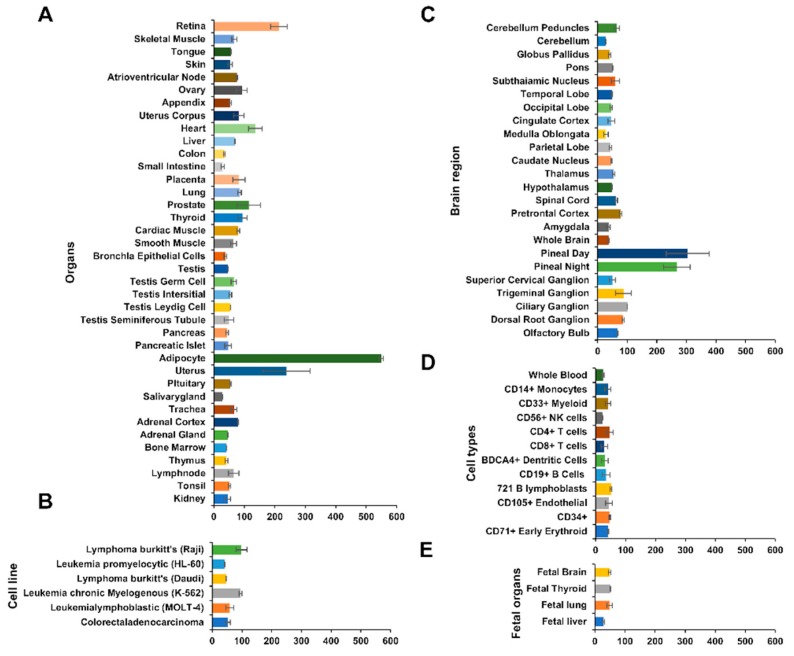
The expression of MFG-E8 in 38 normal tissues, 6 cell lines, 24 brain regions, 12 cell types and 4 fetal organs. Data were derived from BioGPS (Dataset: GeneAtlas U133A, gcrma; http://biogps.org/#goto=genereport&id=4240). The human MFG-E8 mRNA expression in 38 normal tissues (**A**); 6 cell lines (**B**); 24 brain regions (**C**); 12 cell types, (**D**) and 4 fetal organs (**E**). The median of the *x*-axis is 51.4.

**Figure 3 brainsci-08-00167-f003:**
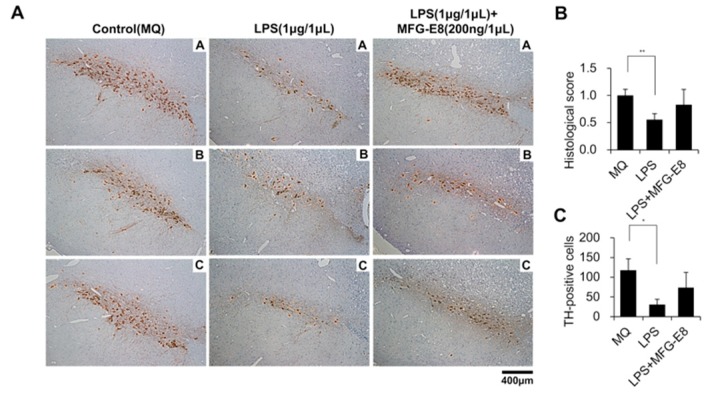
The effect of lipopolysaccharide (LPS) and human MFG-E8 on the dopamine neurons of the substantia nigra. The drug was administered to the rat substantia nigra by manipulation. Brain tissue was harvested at 14 days after administration. A micrograph of the immunohistochemical staining of TH is shown. TH-positive cells, which are stained brown, can be confirmed in the substantia nigra. The left panel shows the tissue sections of 3 rats (**A**–**C**) treated with control: MQ (1 μL). The middle panel shows the tissue sections of 3 rats (**A**–**C**) treated with LPS (1 μg/1 μL). The right panel shows the tissue sections of 3 rats (**A**–**C**) treated with LPS (1 μg/1 μL) + human MFG-E8 (200 ng/1 μL). Scale bar 400 μm (× 100) (**A**). The TH-positive ratio of Figure 3A was calculated with the average value of the control group as 1 (** *p* < 0.01, *n* = 3) (**B**). The number of TH-positive cells in Figure 3A was counted (* *p* < 0.05, *n* = 3) (**C**).

**Figure 4 brainsci-08-00167-f004:**
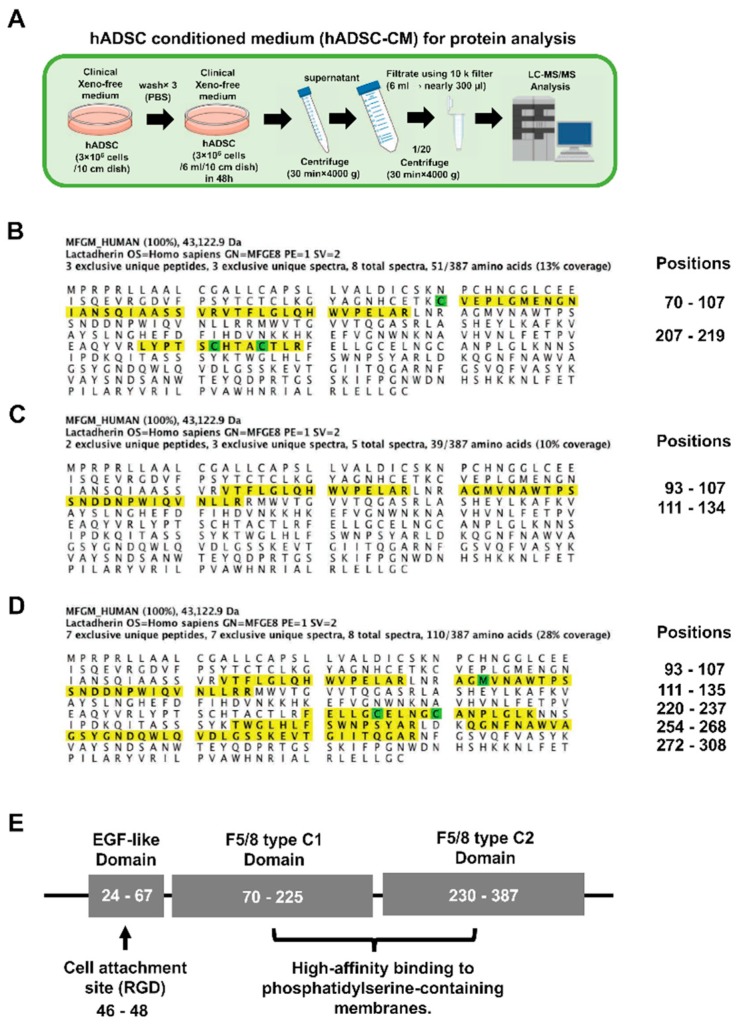
Illustration of the preparation of conditional medium for hADSC and the detection of the peptide sequence in the full length of the protein amino acid sequence is shown. The procedure for preparing the hADSC-CM concentrate for the LC–MS/MS analysis (**A**); The amino acid sequence of MFG-E8 detected in the hADSC-CM sample by the LC–MS/MS analysis is indicated with a yellow marker. The positions of the detected peptides are shown on the right side of the panel. The green marker of the amino acid sequence indicates the modified (C; Carbamidomethyl) amino acid base (**B**); The amino acid sequence of MFG-E8, which was detected in the hADSC (CDM) sample by the LC–MS/MS analysis, is indicated with a yellow marker. The positions of the detected peptides are shown on the right side of the panel (**C**); The amino acid sequence of MFG-E8, which was detected in the hADSC (DMEM-10% FBS) sample by the LC–MS/MS analysis, is indicated with a yellow marker. The positions of the detected peptides are shown on the right side of the panel. The green marker of the amino acid sequence indicates the modified (C; Carbamidomethyl, M; Oxidation) amino acid base (**D**); Illustration of the domain site of human MFG-E8 protein (**E**).

**Figure 5 brainsci-08-00167-f005:**
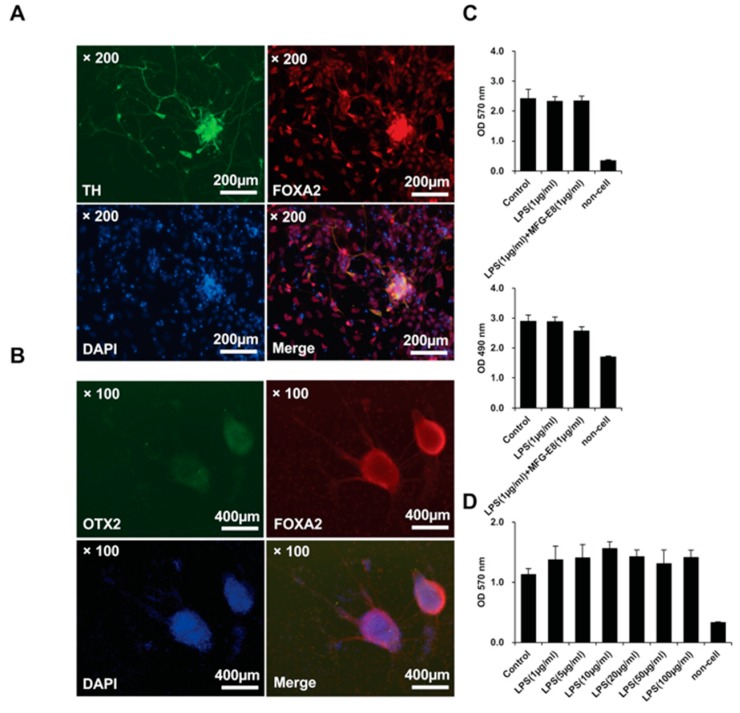
The effect of LPS and human MFG-E8 on the viability of iPSC-derived dopamine neurons. Immunohistochemical staining of TH (top, left panel) and FOXA2 (top, right panel) on iPSC-derived dopamine neurons. The lower panels show 4′6-diamidino-2-phenylindole (DAPI) staining (bottom, left panel) and a merged image of TH-FOXA2-DAPI staining (bottom, right panel). The images were obtained using an optical microscope. Scale bar = 200 μm. (**A**) Immunohistochemical staining of OTX2 (top, left panel) and FOXA2 (top, right panel) on iPSC-derived dopamine neurons. The lower panels show DAPI staining (bottom, left panel) and a merged image of OTX2-FOXA2-DAPI staining (bottom, right panel). The images were obtained using an optical microscope. Scale bar = 400 μm (**B**). Cell viability assays (MTT assay) of iPSC-derived dopamine neurons after 48 h of culture in the presence or absence of 1 μg/mL LPS and 1 μg/mL human MFG-E8. Each measurement was performed using a microplate reader. The absorbance values (570 nm) are indicated. *n* = 4. The data represent the mean ± SD (**C**, top panel). Cytotoxicity assays (LDH assay) of iPSC-derived dopamine neurons after 48 h of culture in the presence or absence of LPS (1 μg/mL) and MFG-E8 (1 μg/mL). Each measurement was performed using a microplate reader. The absorbance values (490 nm) are indicated. *n* = 4. The data represent the mean ± SD (**C**, lower panel). Cell viability assays (MTT assay) of iPSC-derived dopamine neurons after 48 h of culture in the presence or absence of 1, 5, 10, 20, 50, and 100 μg/mL LPS. Each measurement was performed using a microplate reader. The absorbance values (570 nm) are indicated. *n* = 4. The data represent the mean ± SD (**D**).

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
