# Peer review of "The Healing Effect of Human Milk Fat Globule-EGF Factor 8 Protein (MFG-E8) in A Rat Model of Parkinson’s Disease"

_brainsci, 2018, doi:10.3390/brainsci8090167_

Round 1
Reviewer 1 Report
The manuscript Nakashima et al., aims to demonstrate the role human milk fat globule-EGF factor 8 protein (MFG-E8) in a rat model of Parkinson's disease. The results are new and presented in logical manner. Furthermore, authors need to perform additional experiments to understand the effect of MFG-E8.
1. Figure 3, TH IHC staining pictures are in poor quality. Authors need to replace the new one.
2. Figure 5, ICC staining on iPSC culture is not clear.
Author Response
Assist. Prof. Alicia M. Pickrell
School of Neuroscience, Virginia Polytechnic and State University,
Blacksburg, VA 24061, USA
Guest Editor, Brain Sciences
August 23, 2018
Dear Assist. Prof. Pickrell,
We appreciate the thoughtful comments of the reviewers. We have carefully revised our paper in line with those comments, and hope that our newly revised manuscript is now suitable for publication in Brain Sciences.
Respectfully yours,
Yoshiki Nakashima, DDS, PhD.
Department of Regenerative Medicine, Graduate School of Medicine
University of the Ryukyus, 207 Uehara, Nishihara, Okinawa 903-0215, Japan.
Tel: +81-98-895-1696; Fax: +81-98-895-1684; E-mail: nakasima@med.u-ryukyu.ac.jp
RESPONSE TO REVIEWER 1
The manuscript Nakashima et al., aims to demonstrate the role human milk fat globule-EGF factor 8 protein (MFG-E8) in a rat model of Parkinson's disease.
The results are new and presented in logical manner. Furthermore, authors need to perform additional experiments to understand the effect of MFG-E8.
1. Figure 3, TH IHC staining pictures are in poor quality. Authors need to replace the new one.
2. Figure 5, ICC staining on iPSC culture is not clear.
Response: We apologize that the photos in Figures 3 and 5 were blurred and did not meet the publication quality. The images of insufficient quality have now been altered to satisfy the publication quality demanded by the journal (1200 dpi).

Reviewer 2 Report
Dear Authors:
Authors summarized their well-planned study and results in to a manuscript. Authors studied the effects of MFG-E8 on LPS-induced PD rat models. Based on their results authors conclude that MFG-E8 may have therapeutic effects in PD patients. They further claim that, it might also inhibit the microglial activity after the transplantation of dopaminergic neurons derived from hiPSCs.
The results were summed up as a nice manuscript. Congratulations! However, there are few “minor typos and/or comments” to be considered that I recommended, before the manuscript is “accepted” for a publication. Please find below the suggestions, at the end of this message.
Thanks
Comments to the Authors:
1) please have a space between the last word and the [citation number] (eg: page 1; line: 31: change to population [1]), through-out the manuscript!
Introduction
2) page 1; line 41: delete (,) between “PET [4,5].
3) page 2; line: 44; change to “…..suggested to play….”
4) page 2; line: 48: change to “increases”
Materials and Methods
5) page 3; line 125: replace “ad” with “and”
6) page 3; line 127: replace “it” with “to”
7) page 3; line 128: replace “should be” with “were”
8) page 3; line 134: change to “…stopped with forceps.”
Results
9) page 5: line 233: replace “ware” with “were”
10) Figure legends: delete space between the “heading/title” of the figure legend and the rest of the figure legend text, explaining the figure!
If you want to differentiate them, instead of that extra space, please write the “heading/title” of the legend in bold and the rest in plain normal text. This applies to all the figure legends in the manuscript.
11) page 7; line 271: replace “on” with “in”
12) page 9; line 316: replace “Two milliliters” with “2ml”
13) page 12; line 418: delete extra “The” at the beginning of the sentence
14) page 13; line 429-431: what did you mean? If you had the intention to have “subheadings”, why did you not insert them already! OR are you just making a suggestion?
Discussion
15) page 13; line 470: change to “Parkinson’s rat models”
16) page 14; line 482-484: please rephrase the entire sentence.
All the best
Thank you
Author Response
Assist. Prof. Alicia M. Pickrell
School of Neuroscience, Virginia Polytechnic and State University,
Blacksburg, VA 24061, USA
Guest Editor, Brain Sciences
August 23, 2018
Dear Assist. Prof. Pickrell,
We appreciate the thoughtful comments of the reviewers. We have carefully revised our paper in line with those comments, and hope that our newly revised manuscript is now suitable for publication in Brain Sciences.
Respectfully yours,
Yoshiki Nakashima, DDS, PhD.
Department of Regenerative Medicine, Graduate School of Medicine
University of the Ryukyus, 207 Uehara, Nishihara, Okinawa 903-0215, Japan.
Tel: +81-98-895-1696; Fax: +81-98-895-1684; E-mail: nakasima@med.u-ryukyu.ac.jp
RESPONSE TO REVIEWER 2
Authors summarized their well-planned study and results in to a manuscript. Authors studied the effects of MFG-E8 on LPS-induced PD rat models. Based on their results authors conclude that MFG-E8 may have therapeutic effects in PD patients. They further claim that, it might also inhibit the microglial activity after the transplantation of dopaminergic neurons derived from hiPSCs.
The results were summed up as a nice manuscript. Congratulations! However, there are few “minor typos and/or comments” to be considered that I recommended, before the manuscript is “accepted” for a publication. Please find below the suggestions, at the end of this message.
Response: We apologize for the many typographical errors in our text, which has now been revised.
Comments to the Authors:
1) please have a space between the last word and the [citation number] (eg: page 1; line: 31: change to population [1]), through-out the manuscript!
Response: The text has now been modified according to the reviewer’s comments.
Introduction
2) page 1; line 41: delete (,) between “PET [4,5].
3) page 2; line: 44; change to “…..suggested to play….”
4) page 2; line: 48: change to “increases”
Response: The text has now been modified according to the reviewer’s comments.
Materials and Methods
5) page 3; line 125: replace “ad” with “and”
6) page 3; line 127: replace “it” with “to”
7) page 3; line 128: replace “should be” with “were”
8) page 3; line 134: change to “…stopped with forceps.”
Response: The text has now been modified according to the reviewer’s comments.
Results
9) page 5: line 233: replace “ware” with “were”
10) Figure legends: delete space between the “heading/title” of the figure legend and the rest of the figure legend text, explaining the figure!
If you want to differentiate them, instead of that extra space, please write the “heading/title” of the legend in bold and the rest in plain normal text. This applies to all the figure legends in the manuscript.
11) page 7; line 271: replace “on” with “in”
12) page 9; line 316: replace “Two milliliters” with “2ml”
13) page 12; line 418: delete extra “The” at the beginning of the sentence
Response: The text has now been modified according to the reviewer’s comments.
14) page 13; line 429-431: what did you mean? If you had the intention to have “subheadings”, why did you not insert them already! OR are you just making a suggestion?
Response: We apologize for the unnecessary sentences included in the text and have now removed them to avoid any further confusion.
Discussion
15) page 13; line 470: change to “Parkinson’s rat models”
Response: The text has now been modified according to the reviewer’s comments.
16) page 14; line 482-484: please rephrase the entire sentence.
Response: The text has now been modified according to the reviewer’s comments, as follows:
“However, another hypothesis is that MFG-E8 secretes by microglia by itself—this is reported to be an essential factor for the phagocytosis of microglial neurons in a mouse model of Alzheimer's disease. In multiple studies with mice, MFG-E8 has been reported to be an essential factor for neuronal loss in Alzheimer's disease as a microglial secreted protein.”
